# Comparison of Two Electronic Physical Performance Batteries by Measurement Time and Sarcopenia Classification

**DOI:** 10.3390/s21155147

**Published:** 2021-07-29

**Authors:** Chan Mi Park, Hee-Won Jung, Il-Young Jang, Ji Yeon Baek, Seongjun Yoon, Hyunchul Roh, Eunju Lee

**Affiliations:** 1Asan Medical Center, Division of Geriatrics, Department of Internal Medicine, University of Ulsan College of Medicine, Seoul 05505, Korea; chanmipark.paper@gmail.com (C.M.P.); dreamcatch899@gmail.com (J.Y.B.); eunjulee@amc.seoul.kr (E.L.); 2Harvard T.H. Chan School of Public Health, Boston, MA 02115, USA; 3Dyphi Research Institute, Dyphi Inc., Daejeon 34068, Korea; seongjun@dyphi.com (S.Y.); roh@dyphi.com (H.R.)

**Keywords:** physical performance, diagnosis, sarcopenia, sensor

## Abstract

The Short Physical Performance Battery (SPPB) is a widely accepted test for measuring lower extremity function in older adults. However, there are concerns regarding the examination time required to conduct a complete SPPB consisting of three components (walking speed, chair rise, and standing balance tests) in clinical settings. We aimed to assess specific examination times for each component of the electronic Short Physical Performance Battery (eSPPB) and compare the ability of the original three-component examinations (eSPPB) and a faster, two-component examination without a balance test (electronic Quick Physical Performance Battery, eQPPB) to classify sarcopenia. The study was a retrospective, cross-sectional study which included 124 ambulatory outpatients who underwent physical performance examination at a geriatric clinic of a tertiary, academic hospital in Seoul, Korea, between December 2020 and March 2021. For eSPPB, we used a toolkit containing sensors and software (Dyphi, Daejeon, Korea) developed to measure standing balance, walking speed, and chair rise test results. Component-specific time stamps were used to log the raw data. Duration of balance examination, 5 times sit-to-stand test (5XSST), and walking speed examination were calculated. Sarcopenia was determined using the 2019 Asian Working Group for Sarcopenia (AWGS) guideline. The median age was 78 years (interquartile range, IQR: 73,82) and 77 subjects (62.1%) were female. The total mean eSPPB test time was 124.8 ± 29.0 s (balance test time 61.8 ± 12.3 s, 49.5%; gait speed test time 34.3 ± 11.9 s, 27.5%; and 5XSST time 28.7 ± 19.1 s, 23.0%). The total mean eQPPB test time was 63.0 ± 25.4 s. Based on the AWGS criteria, 34 (27.4%) patient’s results were consistent with sarcopenia. C-statistics for classifying sarcopenia were 0.83 for eSPPB and 0.85 for eQPPB (*p* = 0.264), while eQPPB took 49.5% less measurement time compared with eSPPB. Breakdowns of eSPPB test times were identified. Omitting balance tests may reduce test time without significantly affecting the classifying ability of eSPPB for sarcopenia.

## 1. Introduction

Clinical need to improve physical performance in older adults has been emphasized for appropriate risk prediction and clinical decision making [1,2,3,4,5,6] due to an increase in the number of older adults with frail and multimorbid conditions [7,8]. Among the various measures of physical function, the Short Physical Performance Battery (SPPB), composed of three components—standing balance, walking speed, and 5 times sit-to-stand tests (5XSST)—has been extensively studied for its ability to predict functional decline, hospitalization, and mortality under varying circumstances. It is suggested as a relevant outcome measure for clinical interventions targeting frailty and sarcopenia in older adults [4,9,10,11,12,13].

Although SPPB shows excellent performance in predicting outcomes, one of the barriers to performing SPPB in clinical practice is the time required to educate participants on each task and to subsequently conduct and assess the examination. In an original study, the time required to complete SPPB was reported to be 10 to 15 min [10], while in a later study, the protocol could be completed in less than 5 min [11]. To enable wider adoption of SPPB in routine care for older adults, administrative and logistical facts are needed to establish cost data for physical performance examinations. Consequently, breakdowns of the durations of these individual components are also required. However, studies on specific examination times to complete individual components of SPPB are scarce.

Previously, we developed a computer-based toolkit to assess SPPB (eSPPB) in a semi-automatic manner with sensors and algorithms [14]. Subsequently, we updated the eSPPB program to record synchronized timestamp data from the sensors and computer for the three individual exams. Based on these timestamp data and eSPPB results, we sought to estimate component-specific examination times. Furthermore, we compared the ability of the original three-component examination and the faster two-component examination to classify sarcopenia.

## 2. Materials and Methods

### 2.1. Study Design and Population

For retrospective timestamp analysis of eSPPB, we reviewed clinical records of 165 patients who were examined for physical performance in the geriatric outpatient clinic in a university hospital in Seoul, Korea, between December 2020 and March 2021. From this dataset, 17 duplicate records of patients who underwent eSPPB more than once, 14 records with an abnormally long test time (>4 min for balance test), and 10 individuals without comprehensive geriatric assessments were excluded. Finally, 124 older adults were included in the analysis to assess the clinical prediction ability of a shortened version of SPPB (Quick Physical Performance Battery (eQPPB) in this study) and the original eSPPB. In our clinic, community-dwelling patients who are ambulatory with or without a walking aid were considered for the test, as eSPPB includes assessments for walking and standing. However, we excluded: (1) patients with estimated life expectancy of less than 1 year due to advanced malignancy; (2) patients with uncontrolled organ failure including decompensated heart failure or end-stage renal disease; (3) patients unable to walk without other people’s assistance; and (4) patients with cognitive impairment who could not perform eSPPB tasks according to instructions. The study protocol was reviewed and approved by the Institutional Review Board (IRB) in Seoul, Korea, and the requirement to obtain informed consent was waived for this analysis owing to the retrospective nature of the study.

### 2.2. eSPPB Protocol and Time Stamp Data

For eSPPB, we used a toolkit consisting of sensors and software (Dyphi, Daejeon, Korea) developed to measure standing balance, walking speed, and chair rise test results, as reported previously [14]. Standing balance was measured with a loadcell array detecting 2-dimensional location and weight distribution of an individual foot for three postures: side-by-side, semi-tandem, and tandem stances. Participants were asked to maintain each posture for up to 10 s. A four-meter walking speed with a separate one-meter acceleration section that was not included in the calculation was measured using a 1-dimensional light detection and ranging (LiDAR) sensor that records the distance between the sensor and the participant [15]. For 5XSST, participants were asked to stand up five times as quickly as possible, with arms folded on the chest. Time taken to complete the test was measured using two sensors: a loadcell embedded chair that could measure the weight of the sitting participant every 10 milliseconds, and a small chip LiDAR range sensor that could measure the distance between the buttocks of the patient and the chair.

In this protocol, component-specific timestamps were designed to be logged in the raw data. These timestamps were programmed to synchronize three individual modules with the main computer responsible for switching the sensors of these modules on or off using the Xbee protocol [16] when entering or exiting a specific corresponding mode of the three eSPPB components. The duration of balance, 5XSST results, and walking speed were calculated from these data. Instructions, positioning, and actual measurements for individual components were all performed within these three durations. However, the time taken for initial calibration and data entry (patient number, name, age, and sex) was not included in these durations.

### 2.3. Sarcopenia Assessments

We assessed and determined sarcopenia using the 2019 Asian Working Group for Sarcopenia (AWGS) guideline [17]. The appendicular skeletal muscle mass (ASM) was calculated by adding together the muscle masses of the four extremities, assessed using bioelectrical impedance analysis (InBody S10, InBody, Seoul, Korea). Afterward, ASM was divided by height^2^ (m^2^) to calculate skeletal muscle mass index (SMI). Muscle strength was measured by assessing the grip strength of the dominant hand with a JAMAR hydraulic handgrip dynamometer (Patterson Medical, Warrenville, IL, USA), with the elbow flexed at 90 degrees, in a seated position. Physical performance was assessed by measuring the usual gait speed as per the eSPPB protocol. Individuals with low muscle mass (SMI < 7.0 kg/m^2^ for male and < 5.7 kg/m^2^ for female), low muscle strength (grip strength < 28 kg for male and < 18 kg for female), or low physical performance (gait speed < 1.0 m/s) were considered to have sarcopenia.

### 2.4. Statistical Analysis

Differences in basic characteristics between sexes were analyzed using the *t*-test for continuous variables and the χ^2^ test for categorical variables. Durations of individual tests were compared between patients with and without sarcopenia. We used linear regression analysis to assess correlation between eSPPB and eQPPB. For criterion validity, the area under the curve (AUC) of the receiver of characteristics (ROC) curve of eSPPB and eQPPB classifying sarcopenia were compared. We considered two-sided *p*-values of <0.05 as statistically significant. In this study, all analyses were performed using STATA 16.0 (StataCorp, College Station, TX, USA).

## 3. Results

### 3.1. Baseline Characteristics

The median age was 78 years (interquartile range, IQR: 73,82), and 77 (62.1%) participants were female. Based on the 2019 AWGS guideline, 34 (27.4%) individuals were classified as being in the sarcopenia group, of which 21(27.3%) were female and 13 (27.7%) were male. Clinical and functional characteristics are shown in Table 1. In general, clinical parameters showed no significant differences between males and females. The total eSPPB score trended lower in women (*p* = 0.090), while there was no significant difference in the test times of individual components or the total eSPPB protocol between the two sexes.

### 3.2. Test Times

Figure 1 shows the mean time (s) required for the total SPPB test and the mean time for each of the three components between the sarcopenia and non-sarcopenia groups. The calculated mean test time of eSPPB was 125 s. The mean of the total time taken for SPPB by the non-sarcopenia group (119 s) was lower than that of the sarcopenia group (141 s; *p* = 0.004). In the non-sarcopenia group, the balance test accounted for over 50% (62 s, 52.1%) of the full examination time, and 5XSST required the smallest amount of time in both females (27 s, 22.0%) and males (23 s, 20.4%). After excluding the balance test of eSPPB, the calculated mean test time of eQPPB was 63.0 s, which was 50.4% of the eSPPB mean time.

### 3.3. eSPPB, eQPPB, and Sarcopenia Classification

From linear regression analysis, eSPPB and eQPPB correlated with each other (standardized beta, B = 0.94; R^2^ = 0.878). Corresponding mean eSPPB score by individual eQPPB scores are shown in the Appendix A. Briefly, a QPPB score of 2 corresponded to an eSPPB mean score of 4.5 (SD, 1), 5 to 7 (1.1), and 8 (0.7) to 11.7, respectively.

Upon classifying sarcopenia, the C-statistic was 0.83 (0.75–0.91) for eSPPB and 0.85 (0.77–0.92) for QPPB (*p* = 0.264). The eSPPB score cut-point of ≤9 maximized the sensitivity plus specificity for classifying the sarcopenia group, while a cut-point of ≤5 maximized these measures for QPPB. The sensitivities and specificities for the detection of sarcopenia using the two measures are presented in Table 2.

## 4. Discussion

Our study examined the time taken to complete the total eSPPB test and each of the three components using timestamps recorded during measurements. The time taken to perform the balance test exceeded 60 s for all groups and represented the largest portion of the total time required among the three components of SPPB. Excluding the balance test (eQPPB) did not alter the ability to classify sarcopenia significantly when compared with the examination that included the balance test (eSPPB). The eSPPB score cut-point of ≤9 resulted in maximum ability for classifying sarcopenia, while this cut-point was ≤5 for QPPB.

We learned that the full assessment required approximately 3 min, and the duration could be reduced to less than 2 min after excluding the balance test. Our findings are in line with the previous study by Studenski [11], wherein complete battery testing took less than 5 min, and more than 50% of the healthcare providers felt the testing was acceptable and feasible for a routine assessment in a geriatrics clinic. From our observations, the cost structure of adopting SPPB in clinical practice might be better estimated from an administrative point of view. Additionally, using QPPB in the clinical setting may ease the demand for additional staff and reduce the time required to conduct the test.

Of the three tests in SPPB, including balance, gait speed, and chair stand tests, it remains unclear as to which test is the strongest predictor for sarcopenia. A few studies have reported gait speed to be the strongest predictor for health-related adverse outcomes and disability, while some studies demonstrated that poor performance in any one of three components of the SPPB had similar predictive power for health-related events [9,18,19]. We found no significant additional benefits to completing SPPB over QPPB. Moreover, QPPB could be measured in less time while providing similar prognostic power in classifying sarcopenia.

Across numerous studies, investigators focused on the predictive value of SPPB in various settings for different clinical outcomes [4,20,21]. While previous studies focused on health-related events as an outcome of poor SPPB scores, we aimed to detect sarcopenia—a representative geriatric condition caused by a decrease in muscle mass and function—which is directly associated with various adverse health outcomes [22]. In guidelines from Europe and Asia, SPPB, gait speed, and 5XSST results are considered to be components of sarcopenia diagnosis, with SPPB score regarded as a key clinical outcome for improvements in clinical studies with intervention focusing on sarcopenia and physical frailty [12,13,17,23]. Although clinical relevance, including meaningful clinical change of eQPPB, is not known, in the future, QPPB, as well as SPPB, might be used as clinical outcome measures for intervention studies. Further studies assessing both SPPB and QPPB are warranted.

This study has several limitations. We had no records of participants with a total eSPPB score of 3 or less. The study was performed by analyzing medical records from an outpatient clinic of a single center in Korea, and the generalizability of the study is limited. In our data, manually measured SPPB was unavailable since eSPPB was already in full clinical use, and the test times between manual SPPB and eSPPB could not be compared. With the cross-sectional, retrospective nature of the study, clinical outcomes could not be assessed for eSPPB and eQPPB.

## 5. Conclusions

In conclusion, eSPPB could be completed in about 2 to 3 min in real-world clinical practice. By assessing the individual components of eSPPB, we found that the balance test required the longest time among the three components. In classifying sarcopenia, the two-component eQPPB was not significantly inferior to the three-component eSPPB, while substantially reducing total test time.

## Figures and Tables

**Figure 1 sensors-21-05147-f001:**
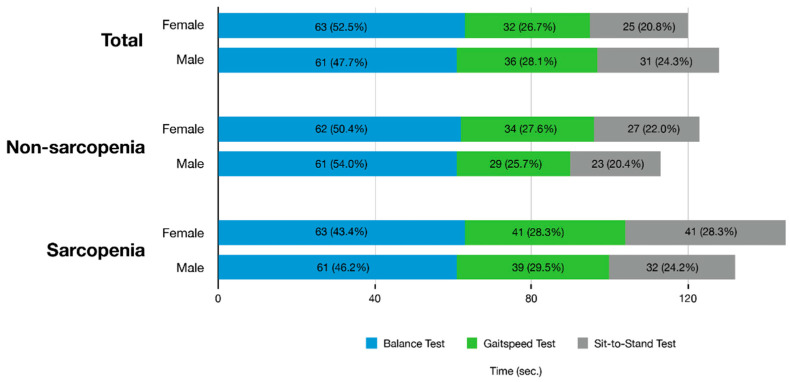
Test time for individual components and the total electronic Short Physical Performance Battery protocol in the study participants with or without sarcopenia.

**Table 1 sensors-21-05147-t001:** Characteristics of the study population.

	Total	Male	Female	*p* Value
Sample size, *n* (%)	124 (100)	47 (37.9)	77 (62.1)	NA
Age, median (IQR)	78 (73,82)	78 (72,83)	78 (73,82)	0.775
BMI kg/m^2^, median (IQR)	24 (22,27)	24 (22,25)	25 (22,27)	0.189
History of DM, *n* (%)	47 (37.9)	15 (31.9)	32 (41.6)	0.283
History of HTN, *n* (%)	76 (61.3)	25 (53.2)	51 (66.2)	0.148
History of stroke, *n* (%)	10 (8.1)	6 (12.8)	4 (5.2)	0.133
History of fall, *n* (%)	18 (14.5)	4 (8.5)	14 (18.2)	0.138
Sarcopenia, *n* (%)	34 (27.4)	13 (27.7)	21 (27.3)	0.963
MMSE, mean (SD)	25.3 (3.9)	25.8 (3.8)	25.0 (4.0)	0.347
SPPB total score, mean (SD)	9.9 (2.4)	10.4 (2.3)	9.7 (2.4)	0.09
SPPB total duration, s, median (IQR)	120 (104,139)	114 (101,132)	122 (105,144)	0.097
Balance test duration, s, median (IQR)	59 (55,68)	59 (54,65)	60 (55,70)	0.278
Gait speed measure duration, s, median (IQR)	32 (27,39)	30 (26,39)	35 (28,40)	0.105
Sit-to-stand test duration, s, median (IQR)	25 (17,36)	22 (16,32)	26 (17,36)	0.253
Polypharmacy, *n* (%)	76 (61.3)	30 (63.8)	46 (59.7)	0.650

DM, diabetes mellitus, HTN, hypertension, IQR, interquartile range; MMSE, mini-mental state examination; SD, standard deviation; SPPB, Short Physical Performance Battery.

**Table 2 sensors-21-05147-t002:** Sensitivity (Sen) and specificity (Spe) of eSPPB and eQPPB scores for the prediction of sarcopenia.

eSPPB Score Cut-Off	eSPPB Score		eQPPB Score Cut-Off	eQPPB Score	
	Sen	Spe		Sen	Spe
≤12	100.0%	0.0%			
≤11	91.2%	54.4%			
≤10	85.3%	66.7%	≤7	100.0%	0.0%
≤9	70.6%	83.3%	≤6	91.2%	65.6%
≤8	64.7%	88.9%	≤5	76.5%	82.2%
≤7	44.1%	91.1%	≤4	55.9%	88.9%
≤6	32.4%	95.6%	≤3	50.0%	93.3%
≤5	20.6%	97.8%	≤2	26.5%	94.4%
≤4	8.9%	100.0%	≤1	11.8%	100.0%
<4	0.0%	100.0%	<1	0.0%	100.0%

eSPPB, electronic Short Physical Performance Battery; eQPPB, electronic Quick Physical Performance Battery.

## Data Availability

The datasets used and/or analyzed during the current study are available from Dr. Hee-Won Jung and Dr. Il-Young Jang on reasonable request.

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
