# Peer review of "Comparison of Two Electronic Physical Performance Batteries by Measurement Time and Sarcopenia Classification"

_sensors, 2021, doi:10.3390/s21155147_

Round 1

Reviewer 1 Report

This paper ultimately compares the three-component eSPPB and the two-component eQPPB for classifying sarcopenia. The article can be potentially very interesting as it provides evidence on the reduction of time in the evaluation related to the classification of sarcopenia, so its contributions can have an impact on the economic, human resources or time level.  The article, although well written, is very brief and on some occasions clarifications are needed to understand the justification of the subject matter. In addition, quotations are found after the "." or "," of the sentence to which they refer, which makes it difficult to know to which statement the quotation in question corresponds. 

Minor points: 

Introduction: 

The introduction is concise, but a justification of why it is important to reduce the evaluation time of the test or the impact it may have would be advisable. For example, with respect to personal resources, economic repercussion on the health system or the area in which the test is developed.
Line 47-49. What was the reason for the decrease in time in this case, and were any tests eliminated? Could we justify why the time was reduced on other occasions? 

Material and methods. 

Line 67. Why was ">4 min" chosen, on the basis of what was this criterion established? If the mean for the total sample is "59 sec", would it not be convenient to lower this criterion to 2 or even 3 minutes, since it would still double the time of the mean?

Statistical analysis 

Line 117. It should be included that differences were made between sexes, as I understand that in Table 1 the "p" value refers to between sexes. If not specified, this may lead to confusion since it is only specified in this paragraph that comparisons were made between patients with and without sarcopenia. 
Line 122. It is stated that the Kappa statistic was used, but it is not specified in the results section. 
In addition, it is not specified that a regression line will be performed nor on the basis of which it was established. 

Results

In relation to the "IQR", this is defined as Q3-Q1, so it is not understood: 
- For example in: "Age, median (IQR)"; the Q2 is 78 years and Q3-Q1=73.82. 
Table 1. It is recommended that the "." instead of the "," for the whole table. 

Discussion 

As has been expressed during the review, it could be enriched by providing specific data on the impact that the decrease in time may have on the evaluation of the different tests. 

Author Response

REVIEWER 1

Comment 1:

This paper ultimately compares the three-component eSPPB and the two-component eQPPB for classifying sarcopenia. The article can be potentially very interesting as it provides evidence on the reduction of time in the evaluation related to the classification of sarcopenia, so its contributions can have an impact on the economic, human resources or time level.  

Response: Thank you. We appreciate the reviewer for these positive comments.

Comment 2:

The article, although well written, is very brief and on some occasions clarifications are needed to understand the justification of the subject matter. In addition, quotations are found after the "." or "," of the sentence to which they refer, which makes it difficult to know to which statement the quotation in question corresponds. 

Response: We appreciate the reviewer for the comments. We carefully edited our quotations to what is clearly referring to.

Comment 3:  

The introduction is concise, but a justification of why it is important to reduce the evaluation time of the test or the impact it may have would be advisable. For example, with respect to personal resources, economic repercussion on the health system or the area in which the test is developed.

Response:  It is important to reduce the evaluation time of the full SPPB exam because one of the major barriers of resources of the exam is time needed to perform in the clinics and whether they are applicable in the real world. In the study by Studensky et al., they raise the same question whether other epidemiological studies that are previously done may not apply to clinical settings and main aim of the study was to determine whether simpler assessment can be used in real-world for certain outcomes. (Studenski S, Perera S, Wallace D, Chandler JM, Duncan PW, Rooney E, Fox M, Guralnik JM. Physical performance measures in the clinical setting. J Am Geriatr Soc. 2003 Mar;51(3):314-22. doi: 10.1046/j.1532-5415.2003.51104.x. PMID: 12588574.)

Therefore, our study findings add values in attempt to determine the most efficient approach in real-world clinical settings. If validity of the exam stays the same with a shortened amount of time for measuring an outcome, many more physicians would actively ascertain SPPB in their outpatient clinics where time and resources are limited in a more efficient way.    

Comment 4:

Line 47-49. What was the reason for the decrease in time in this case, and were any tests eliminated? Could we justify why the time was reduced on other occasions? 

Response: Thank you for making this important point. Both studies, which were cited in the manuscript used all three components for the SPPB examinations. Both studies did not eliminate any of the components of the exam. They were individually conducted studies. The time was reduced not on the same matter of occasion, but it was reduced in different study settings. The later study does not explain their outcomes based on the previous cited study. 

Comment 5:

Line 67. Why was ">4 min" chosen, on the basis of what was this criterion established? If the mean for the total sample is "59 sec", would it not be convenient to lower this criterion to 2 or even 3 minutes, since it would still double the time of the mean?

Response: Thank you for raising this question. We used >4 min as a cut point as an abnormality because there were some subjects lie between 80 -90 percentile which was from 2-3 minutes range.

Comment 6:

Line 117. It should be included that differences were made between sexes, as I understand that in Table 1 the "p" value refers to between sexes. If not specified, this may lead to confusion since it is only specified in this paragraph that comparisons were made between patients with and without sarcopenia.

Response: I am sorry for the confusion and thank you for clarifying. We made a change to the first sentence. 

“Basic characteristics differences between sexes were analyzed using the t-test for continuous variables and the χ2 test for categorical variables.” (Line 144)

Comment 7:

Line 122. It is stated that the Kappa statistic was used, but it is not specified in the results section. 
In addition, it is not specified that a regression line will be performed nor on the basis of which it was established. 

Response:  Thank you for the clarification. With the continuous nature of eSPPB and eQPPB, we used linear regression analysis rather than using kappa statistic, and revised the method part to comply with the final analysis.

            “We used linear regression analysis to assess correlation between eSPPB and eQPPB. For criterion valitdity, Tthe area under the curve (AUC) of the receiver of characteristics (ROC) curve of eSPPB and eQPPB classifying sarcopenia were compared.” (Line 146)

            “Kappa statistic was calculated to assess the agreement between impaired physical performance status determined by eSPPB and eQPPB” -> deleted

Comment 8:

In relation to the "IQR", this is defined as Q3-Q1, so it is not understood: 
- For example in: "Age, median (IQR)"; the Q2 is 78 years and Q3-Q1=73.82. 
Table 1. It is recommended that the "." instead of the "," for the whole table. 

Response: To make a clarification, “IQR”was defined as an interquartile range in our study. It is stated in the footnote of the Table 1. Two numbers in the parenthesis account for (25th percentile, 75th percentile) numbers. The punctuation should remain as “,” in order to represent two separate numbers.

Comment 9:

As has been expressed during the review, it could be enriched by providing specific data on the impact that the decrease in time may have on the evaluation of the different tests. 

Response: I strongly agree with the reviewer’s comment. The best specific data that could be presented as an impact of the decrease in time was the similar predictive ability to measure sarcopenia between the two exams (Table 2.) We believe that c-statistic values display minimal impact of decreasing test protocol in terms of classifying sarcopenia.

Reviewer 2 Report

The paper intends to compare two electronic physical performance batteries on measurement time.

The abstract reflects the paper's content. Though some acronyms need to be defined.

The document is well structured. The literature review is adequate and complete.

The study is well-designed et the result well presented. I appreciate the discussion.

I will strongly recommend accepting the paper in this form.

__________________________________________________________________________________

The manuscript reports a study of a comparative assessment of physical batterie performance with a focus on measurement time and Sarcopenia Classification. Performance batteries are used in eldercare for measuring their extremity sensibility. Though, the authors motivate their work by the time issues faced in measuring the elder extremity functions. Therefore, they conducted a retrospective, cross-sectional comparative study to figure out which component of the measurement impacts the time, where they compared 3 component-based examinations and 2 component-based examinations without a balance test to classify sarcopenia.
The results have been conclusive.

The authors structure the manuscript. The content is easy to read and understandable.
The study method and design are well described and clear.
the introduction presents a clear tableau of the problem statement ánd concisely present the background, I mean the prior work they have done.
The study method section clearly presents the study materials (data), participant sampling,  sarcopenia assessment method, etc. It is easy to follow. The method is followed by a well-structured result section, where the main evidence is presented. The comparison between both examination methods is presented and clearly described.
I appreciate the discussion since this is supported by the literature and clearly analyzes the results.
The paper provides a clear conclusion that summarizes the main study findings.

Despite the quality of the manuscript structure, the presentation, and the clarity of the content, the manuscript presents the following minor weaknesses:
              (I) "complete SPPB consisting of three components in clinical settings." I cannot clearly figure out along the manuscript what these three components are. The authors will be advised to describe the components they are talking about.
              (II) "two component-based examination without a balance test (electronic 16 Quick Physical Performance Battery, eQPPB) to classify sarcopenia" same as (I)
              (III) "complete SPPB consisting" the acronym is not defined in the abstract by the first use. 

Author Response

REVIEWER 2

Comment 1:

The paper intends to compare two electronic physical performance batteries on measurement time. The abstract reflects the paper's content. Though some acronyms need to be defined. The document is well structured. The literature review is adequate and complete. The study is well-designed et the result well presented. I appreciate the discussion.I will strongly recommend accepting the paper in this form.

The manuscript reports a study of a comparative assessment of physical batterie performance with a focus on measurement time and Sarcopenia Classification. Performance batteries are used in eldercare for measuring their extremity sensibility. Though, the authors motivate their work by the time issues faced in measuring the elder extremity functions. Therefore, they conducted a retrospective, cross-sectional comparative study to figure out which component of the measurement impacts the time, where they compared 3 component-based examinations and 2 component-based examinations without a balance test to classify sarcopenia.
The results have been conclusive.

The authors structure the manuscript. The content is easy to read and understandable.
The study method and design are well described and clear.
the introduction presents a clear tableau of the problem statement ánd concisely present the background, I mean the prior work they have done.
The study method section clearly presents the study materials (data), participant sampling,  sarcopenia assessment method, etc. It is easy to follow. The method is followed by a well-structured result section, where the main evidence is presented. The comparison between both examination methods is presented and clearly described.
I appreciate the discussion since this is supported by the literature and clearly analyzes the results.
The paper provides a clear conclusion that summarizes the main study findings.

Response: We appreciate very much the reviewer for these positive comments. All suggestions by the reviewer will be answered thoroughly. 

Comment 2:

Despite the quality of the manuscript structure, the presentation, and the clarity of the content, the manuscript presents the following minor weaknesses:
              (I) "complete SPPB consisting of three components in clinical settings." I cannot clearly figure out along the manuscript what these three components are. The authors will be advised to describe the components they are talking about.

Response: Sorry for the confusion and this issue may have been drawn from an assumption. “Short Physical Performance Battery (SPPB) test consisting of three components” means the original version of SPPB that includes all three of tests for a full examination: walking speed, chair rise test and the standing balance.  We added a parenthesis after the phrase and included what those three tests are.

Comment 3:
              (II) "two component-based examination without a balance test (electronic 16 Quick Physical Performance Battery, eQPPB) to classify sarcopenia" same as (I)

Reponse: Aligning with the previous comment, two component-based examination is the “new” version we created which excludes balance test out of the three.

Comment 4:
              (III) "complete SPPB consisting" the acronym is not defined in the abstract by the first use. 

 Response: Sorry for this mistake. The acronym has been added in the very first line of the abstract. Thank you for making this point.